# OpenReview forum: "SFW sampling for diffusion models via external conditioning"
_ICLR.cc/2025/Conference — ICLR 2025 Conference Withdrawn Submission_

### Official Review · Reviewer_UhVu · 2024-11-01

**Soundness:** 3
**Presentation:** 2
**Contribution:** 1
**Rating:** 3
**Confidence:** 4

**Summary:**

This paper studies an important problem of safe-for-work sampling from diffusion models to avoid generating explicit content. To this end, they propose a novel approach to leverage CLIP-based embeddings of harmful concepts and guide the sampling of diffusion models away from this embedding space. They specifically use manifold preserving guidance and only apply it when the classifier is confident enough (i.e., its probability is above a threshold). The proposed method is then compared against baselines to find competitive performance with baselines in explicit content detection and quality and prompt-concordance of the generated image.

**Strengths:**

- The paper is easy to follow and self-contained.
- The experimental design and the proposed method are sound and make sense to me.
- Sections 4 and 2.2 are quite useful in understanding the generalizability of the method.
- Aesthetic degradation experiment is important for the work and I commend the authors for doing that.

**Weaknesses:**

- The empirical comparison with traditional guidance-based methods is missing. It has been discussed that this is different from classifier-based guidance but no empirical evidence is provided to show that the proposed method is better.
- A related technique of negative prompts is not compared or discussed:
  - https://minimaxir.com/2022/11/stable-diffusion-negative-prompt/
- In almost all cases and especially in Table 1, ESD outperforms the proposed method and this negative result is not adequately discussed. This is strange since it seems that the proposed method is not working.
- The only positive result is when the prompts are already safe (Table 3), which seems to indicate that the proposed method is beneficial due to its soft ways of conditioning and steering away from certain regions in an embedding space.
- Novelty of the method is also limited since it uses off-the-shelf classifiers and manifold-preserving guidance approach along with a simple thresholding mechanism on the confidence.
- In the conclusion, the authors note that explicit conditioning is important to uphold the bias of existing diffusion models. This is interesting but due to the lack of any empirical evidence to support this claim, it is hard to establish that explicit conditioning is indeed the way forward. Thus, if this is a claim that the authors want to establish, some results would be really useful and relevant.
- Other important related work are missing from experiments and/or discussion.
  - Hong, Seunghoo, Juhun Lee, and Simon S. Woo. "All but One: Surgical Concept Erasing with Model Preservation in Text-to-Image Diffusion Models." Proceedings of the AAAI Conference on Artificial Intelligence. Vol. 38. No. 19. 2024.
  - Lyu, Mengyao, et al. "One-dimensional Adapter to Rule Them All: Concepts Diffusion Models and Erasing Applications." Proceedings of the IEEE/CVF Conference on Computer Vision and Pattern Recognition. 2024.
  - Pham, Minh, et al. "Circumventing concept erasure methods for text-to-image generative models." The Twelfth International Conference on Learning Representations. 2023.

**Questions:**

Please see above weaknesses

---

### Official Review · Reviewer_dJqX · 2024-11-03

**Soundness:** 2
**Presentation:** 2
**Contribution:** 2
**Rating:** 3
**Confidence:** 4

**Summary:**

This paper addresses the challenge of minimizing the generation of undesired, harmful content in Score-Based Models. The distribution of such harmful samples is represented by a probability density $p_h$.

The authors aim to estimate the conditional probability $p_h(x_0 | x_t, t)$ to identify how likely a sample $x_t$ will produce harmful content at state $x_0$ in the generative process. They propose searching for candidate samples $x_{t-1}$ in the vicinity of $x_t$ that satisfy two criteria: i) they are valid samples, and ii) they yield low density $p_h(x_0 | x_t, t)$.

To implement this, the authors introduce a Conditional Trajectory Correction (CTC) mechanism. The proposed CTC evaluates the NSFW probability of each clean point prediction and applies a corrective adjustment if this probability exceeds a predetermined threshold $ \eta > 0 $. This approach aims to effectively reduce the generation of harmful content while maintaining the integrity of valid sample generation.

**Strengths:**

The authors propose an interesting approach to generating SFW samples with SBMs, leveraging the solver's working principles for the reverse process. Herein lies the potential to extend the method to a broader range of applications beyond content moderation. By directly influencing the sampling trajectory based on explicit probability thresholds, this method could adapt to domain-specific needs where content control is essential, such as ensuring ethical content in SBM-driven artwork or meeting regulatory standards in automated media production. Furthermore, the Conditional Trajectory Correction (CTC) is theoretically grounded.

**Weaknesses:**

One of the primary limitations of this paper lies in its reliance on an implicitly defined $p_h$ using trained classifiers. This approach introduces a dependence on the quality and generalizability of these classifiers, which by the authors' admission may not consistently or accurately capture all harmful content, especially if the classifiers have limited scope or biased training data.

Additionally, the method requires a predefined set of harmful concepts, $\mathcal{C}$, to guide content filtering. In this study, the largest considered set was $\mathcal{C}$ = {violence, nudity, NSFW, harmful}. However, this may not be comprehensive enough to capture the full range of potentially harmful content across various contexts, particularly in domains where nuanced or emerging types of harmful content need to be addressed. This limitation in scope could reduce the method’s effectiveness in broader applications.

Finally, the reported results show little to no improvement over Erasing Stable Diffusion (ESD), a competing approach. This raises questions about the practical advantages of the proposed Conditional Trajectory Correction (CTC) method. Given its performance, it remains unclear if the added complexity of CTC justifies its use over established methods like ESD, especially in settings where computational efficiency and ease of implementation are critical considerations.

**Questions:**

In general, I'm willing to raise my score, if an area can be explored where the proposed method yields a significant performance improvement. Regarding the aforementioned weaknesses, I pose the following questions:

1. $p_h$ is defined implicitly using trained classifiers. If the probability density $p_h$ of harmful samples is reliant on classifiers, what are the benefits of using your approach, rather than removing harmful descriptions from the prompt?

2. Have you considered using a more elaborate set of concepts $\mathcal{C}$ for harmful content detection? How would expanding $\mathcal{C}$ impact the robustness and generalizability of the CTC approach?

3. In addition to question 2: How does the cardinality $|\mathcal{C}|$ affect the computational cost of the proposed CTC method and how does it affect performance? Can you provide an upper bound on computational cost or some time measurements for inference, especially for increasing $|\mathcal{C}|$?

4. The results indicate limited improvement over Erasing Stable Diffusion (ESD). Could you elaborate on any advantages of CTC in scenarios or datasets where it may outperform ESD? Were there specific cases in your experiments where CTC provided a distinct advantage?

5. Given that the effectiveness of CTC is influenced by the threshold $\eta$. The hyper-parameter selection of $\eta=0.23$ was hand-selected based on a grid search. How sensitive is the performance of your approach to different values $\eta \in (0,0.23)$?

6. Could you clarify any potential limitations in terms of scalability or generalization of the proposed CTC method? Specifically, are there specific concepts $\mathcal{C}$ where the approach struggles, and what could be done to address these limitations?

---

### Official Review · Reviewer_9M4e · 2024-11-04

**Soundness:** 2
**Presentation:** 3
**Contribution:** 2
**Rating:** 5
**Confidence:** 3

**Summary:**

This paper presents a method to make diffusion models safer by reducing the likelihood of generating NSFW content. The authors leverage an external signal from CLIP to guide the generation process, steering it away from harmful or explicit images. With the introduction of a Conditional Trajectory Correction step, the model subtly aligns generated content to safe categories without compromising image quality.

This flexible approach allows users to define what qualifies as harmful content, with customizable categories suited to different contexts. Experiments on Stable Diffusion demonstrate that the method reduces NSFW content with minimal impact on image quality, while a prompt-alignment metric assesses faithfulness to the user’s intent. Overall, this method provides a practical step toward safer generative models aligned with user-defined standards.

**Strengths:**

- **Significance:** The studied problem is very important in practice.

- **Readability:** This paper avoids overly complex writing, which enhances comprehension for a broad audience.

- **Minimal Impact on Image Quality:** The proposed method has a minor impact on the quality of benign samples, maintaining the aesthetic appeal of images that do not require correction.

**Weaknesses:**

- **Moderate Impact:** The experimental results show a limited reduction in NSFW content. The evaluation metrics are somewhat narrow; incorporating widely adopted metrics like FID or Inception Score could provide a more comprehensive view of how the method impacts visual quality and generation fidelity.

- **Computational Overhead:** The Conditional Trajectory Correction step likely increases computational and time costs due to added gradient estimations at each inference step. The authors do not discuss this in detail, and it would strengthen the paper to include a comparison of inference times with and without this step. Additionally, while the Conditional Diffusion Trajectory Correction adapts the manifold-preserving guidance from He et al., 2024, this adaptation may not represent a substantial advancement over existing methods.

- **Parameter Sensitivity:** The model’s effectiveness depends on tuning parameters like the harmfulness threshold and guidance strength, which may vary by application. This sensitivity could hinder usability, as careful parameter adjustments might be needed for different scenarios. Including guidance or analysis on parameter selection would enhance practicality.

**Questions:**

- **Complementing Direct Classification:** Could the authors discuss how their method complements or improves on using a direct classifier to detect NSFW content? Specifically, are there notable trade-offs in generation fidelity when comparing this approach to post-generation filtering?

- **Stable Diffusion Configurations:** Stable Diffusion has various versions and parameter settings that might impact results. Could the authors clarify which versions (and parameter settings) were used in their experiments? A sensitivity analysis would improve replicability.

---

### Official Review · Reviewer_PWtK · 2024-11-06

**Soundness:** 3
**Presentation:** 3
**Contribution:** 2
**Rating:** 5
**Confidence:** 4

**Summary:**

The paper proposes a safe-for-work (SFW) sampling method for diffusion models to prevent the generation of not-safe-for-work (NSFW) content. The key innovation is using external multimodal models (specifically CLIP) as a source of conditioning to guide samples away from undesired regions in the ambient space.

**Strengths:**

-  The paper presents a interesting approach to address the issue of NSFW content generation in SBMs by utilizing external conditioning signals.
- The SFW sampler allows for user-defined NSFW classes, making it adaptable to different settings and applications.

**Weaknesses:**

- No ablation studies on different choices of external multimodal models besides CLIP.
- The proposed method is a direct application of manifold-preserving sampler.
- The quantitative results in experiments are not convincing. The proposed method does not show better performance.

**Questions:**

- What is the computational overhead of the proposed method compared to standard sampling?
- Have you explored using other vision-language models besides CLIP for external conditioning?

---

### Note · Authors · 2024-12-01

I have read and agree with the venue's withdrawal policy on behalf of myself and my co-authors.